# Selective Adsorption of Aqueous Diclofenac Sodium, Naproxen Sodium, and Ibuprofen Using a Stable Fe_3_O_4_–FeBTC Metal–Organic Framework

**DOI:** 10.3390/ma14092293

**Published:** 2021-04-28

**Authors:** Aldo Arturo Castañeda Ramírez, Elizabeth Rojas García, Ricardo López Medina, José L. Contreras Larios, Raúl Suárez Parra, Ana Marisela Maubert Franco

**Affiliations:** 1Materials Chemistry, Basic Sciences, Metropolitan Autonomous University-Azcapotzalco, Mexico City 02200, Mexico; amf@azc.uam.mx; 2Process Engineering and Hydraulics, Basic Sciences, Metropolitan Autonomous University-Iztapalapa, Mexico City 09340, Mexico; erg@xanum.uam.mx; 3Energy, Basic Sciences, Metropolitan Autonomous University-Azcapotzalco, Mexico City 02200, Mexico; rilome@correo.azc.uam.mx (R.L.M.); jlcl@azc.uam.mx (J.L.C.L.); 4Institute of Renewable Energies, National Autonomous University of Mexico, Morelos 62580, Mexico; rsp@ier.unam.mx

**Keywords:** metal–organic frameworks (MOF), nonsteroidal anti-inflammatory drugs (NSAIDs), FeBTC, magnetite

## Abstract

The FeBTC metal–organic framework (MOF) incorporated with magnetite is proposed as a novel material to solve water contamination with last generation pollutants. The material was synthesized by in situ solvothermal methods, and Fe_3_O_4_ nanoparticles were added during FeBTC MOF synthesis and used in drug adsorption. X-ray diffraction (XRD), Fourier-transform infrared spectroscopy (FTIR), and Raman spectroscopy characterized the materials, with N_2_-physisorption at 77 K. Pseudo-second-order kinetic and Freundlich models were used to describe the adsorption process. The thermodynamic study revealed that the adsorption of three drugs was a feasible, spontaneous exothermic process. The incorporation of magnetite nanoparticles in the FeBTC increased the adsorption capacity of pristine FeBTC. The Fe_3_O_4_–FeBTC material showed a maximum adsorption capacity for diclofenac sodium (DCF), then by ibuprofen (IB), and to a lesser extent by naproxen sodium (NS). Additionally, hybridization of the FeBTC with magnetite nanoparticles reinforced the most vulnerable part of the MOF, increasing the stability of its thermal and aqueous media. The electrostatic interaction, H-bonding, and interactions in the open-metal sites played vital roles in the drug adsorption. The sites’ competition in the multicomponent mixture’s adsorption showed selective adsorption (DCF) and (NS). This work shows how superficial modification with a low-surface-area MOF can achieve significant adsorption results in water pollutants.

## 1. Introduction

Many industrial pollutants can be found in wastewater. The industrial sector manufactures different chemical reagents to support factories or supply the market with the necessary nutritional, residential, farming, livestock, pharmaceutical, and personal products. Simultaneously, along with these products, the manufacturing sector generates great quantities of chemical waste that are released into the air, soil, and water. Sooner or later, most of the pollutants make their way into rivers, lakes, and seas. Moreover, people and animals transform these products through use, generating residual inorganic and organic compounds from detergents, pharmaceuticals, and personal care products (PPCPs) [1]. Indeed, these pollutants constitute a significant threat to flora and fauna and public health. Therefore, to supply populations with necessary drinking water, it is imperative to find effective cleaning wastewater procedures [2]. However, in the case of wastewater contaminated with nonsteroidal anti-inflammatory drugs (NSAIDs), the cleaning treatment is complex and expensive due to the high concentrations of drugs [3]. When this wastewater is mixed with inorganic pollutants (cation metals, fluorides, nitrates, and phosphates), the cleaning treatment becomes more difficult [4].

Metal–organic frameworks (MOF) are materials with metal aggregates significantly bound by organic ligands containing potential adsorption sites; this feature generates high porosity, and consequently, a highly specific surface [5,6]. Ligands are typically bidentate, tridentate, or tetradentate organics such as benzene dicarboxylate (BDC) or benzenetricarboxylate (BTC) [7]. FeBTC (iron–benzenetricarboxylate) is a MOF with the molecular formula C_9_H_3_FeO_6_, commercially known as Basolite F300. Although its chemical composition is well known, it shows poor crystallinity, and its structure is unknown [8]. Efforts to elucidate the structure suggest an octahedral metal cluster formed of iron trimers connected to benzene tricarboxylate ligands [9]. Additionally, this MOF shows unsaturated vertices that interact with water molecules and open metal sites (Fe^2+/3+^), high water stability, drug-compatible pore size (2 nm diameter), many structural defects, etc. [10]. Based on these characteristics, this material could be an excellent candidate for the adsorption of drugs in water.

PPCPs such as diclofenac sodium (DCF), naproxen sodium (NS), and ibuprofen (IB) (Figure 1) have been studied as contaminants to be removed by MOFs because they are among the most abundant in polluted waters. The MOFs MIL-101 (Cr) and MIL-100 (Fe), studied by Jhung et al. [11], proved to be good candidates to remove naproxen and clofibric acid. The authors deduced that electrostatic interaction and stereo porous selectivity dominate the adsorption procedure by analyzing the pollutant capture process. Later, Jhung et al. [12] studied the adsorption of the same pollutants on MIL-101-Cr functionalized with acidic (−SO_3_H) and alkaline (−NH_2_) groups to determine their influences in the aqueous system.

On the other hand, magnetite has proven to be an excellent candidate for removing pollutants in an aqueous medium [13]. Its ability to incorporate quite well with other materials [14,15,16] increases adsorption capacity and recovery in aqueous medium and thermal stability [17]. Although MOFs included with magnetite have already been studied, few have been applied to remove contaminants in aqueous media [14,18]. Giesy et al. [19] reported one of the first works to incorporate magnetite in MOFs using HKUST-1 included with magnetic material to remove colorants from water, showing a clear increase in the removal capacity of methylene blue. Chen et al. [20] used a MIL-100 (Fe) MOF incorporated with magnetite to remove rhodamine B from the medium, highlighting the effective reuse of the material, the improvement of the removal capacity of the dye, and the stacking of pi bonds and electrostatic interactions as the main possible interactions. Hou et al. [21] used the Freundlich adsorption model and pseudo-second-order kinetic model to describe the adsorption process of MIL-100 (Fe) with magnetite for use with meloxicam and naproxen. Shortly after, the same research group [22], using the same material, showed that photodegradation of pollutants [6,21] such as diclofenac and other drugs was possible.

Although several studies use MOFs incorporated with magnetite, most of them are limited to MIL-100 and other MOFs with highly specific areas. This study investigated three drugs (ibuprofen, naproxen sodium, and diclofenac sodium) commonly used without a prescription, making them the most prevalent and the most important in waste management. We also demonstrate the benefits of a MOF with a surface area of 217 m^2^/g capable of obtaining superior adsorption capacity, showing preferential adsorption sites through Raman and FTIR characterization after the adsorption process.

In the present study, we proposed magnetite nanoparticles as a good interaction material with the FeBTC MOF because magnetite incorporated into various materials has been shown to improve the capacity to remove drugs through electrostatic interactions with their functional groups [23,24,25]. It modified the surface properties of the FeBTC and increased its interaction energy with diclofenac, naproxen, and ibuprofen. Before and after the material characterization was carried out to identify the drugs’ potential adsorption sites in the MOF, the three drugs’ adsorption isotherms and kinetics were studied. Besides, the thermodynamic parameters were calculated for the adsorption of the compounds on the adsorbent.

## 2. Materials and Methods

### 2.1. Reagents

Sodium acetate anhydrous (CH_3_COONa, purity ≥ 99%), ethylene glycol anhydrous (HOCH_2_CH_2_OH, purity ≥ 99%), anhydrous iron nitrate (III) (Fe(NO_3_)_3_·9H_2_O), trimesic acid (C_6_H_3_(CO_2_H)_3_, purity ≥ 95%), N, N-dimethylformamide anhydrous (HCON(CH_3_)_2_, purity = 99.8%), diclofenac sodium salt (C_14_H_10_C_l2_NNaO_2_, purity ≥ 98%), naproxen sodium (C_14_H_13_NaO_3_, purity ≥ 98%), ibuprofen (C_13_H_18_O_2_, purity ≥ 98%) all purchased from Sigma–Aldrich(St. Louis, MO, USA), and iron (III) chloride hexahydrate (FeCl_3_·6H_2_O, Reag. Ph Eur) purchased from Emsure (Darmstadt, Germany).

### 2.2. FeBTC Synthesis

The FeBTC MOF was synthesized using the solvothermal method, as used by Rojas et al. [26]. Iron nitrate hexahydrate (8.7 mmol) and trimesic acid (H_3_BTC, 8.3 mmol) were added to 30 mL of N, N-dimethylformamide (DMF) and sonicated in an ultrasound bath for 5 min. Subsequently, we added 30 mL of ethanol and 30 mL of deionized water to the mixture, which was sonicated for 35 more minutes to ensure complete dissolution of the components. Finally, the mixture was subjected to heating in a sand bath for 24 h at 358 K [27].

### 2.3. Fe_3_O_4_ (Magnetite) Synthesis

Magnetite was obtained according to the post-synthesis method of Liu et al. [20] with modifications. Briefly, iron (III) chloride hexahydrate (5.06 mmol) and anhydrous sodium acetate (23.65 mmol) were added to 50 mL of ethylene glycol (50 mL). The previous dissolution was introduced in a steel autoclave with Teflon liner (100 mL). The autoclave was placed at 473 K for 10 h. The solid obtained was washed with ethanol and water several times to remove impurities. Finally, the magnetite was dried in an oven at 353 K for 12 h.

### 2.4. Synthesis of the Fe_3_O_4_–FeBTC Composite

Synthesis of the Fe_3_O_4_–FeBTC composite was based on the one-step hydrothermal/solvothermal method reported by Jiang et al. [28]. An exact amount of magnetite (incorporation at 2% and 8%) was also studied. Still, in this study, only the best result is reported: 5% wt. was added to 30 mL of ethanol and ultrasonicated to ensure complete dissolution of the components (solution A). Solution B was dissolved in DMF with trimesic acid precursors (7.88 mmol) and iron nitrate hexahydrate (8.26 mmol). Subsequently, both solutions were mixed and placed into an ultrasound bath for 5 min, then 10 mL of deionized water was added, and the solutions were sonicated for another 30 min. The previous solution was kept under vigorous stirring at 358 K for 24 h. Finally, the composite was dried in an oven at 353 K for 12 h.

In this work, we used non-functionalized magnetite nanoparticles. It has been proven that simple incorporation (non-functionalized in situ) is a fast and efficient way to obtain magnetic materials composites Fe_3_O_4_–MOFs [29] with good dispersion and synergy [20,30]. A significant advantage of not functionalizing the magnetite lies in preserving the magnetic properties since they are crucial in the material’s recyclability. That is, these properties make it easier to recover the adsorbent by applying an external magnetic field.

### 2.5. Activation of the Materials

For activation of FeBTC and Fe_3_O_4_–FeBTC to free up their pores following synthesis, each component was subjected to a 3-step procedure. First, they were washed with methanol and then filtered with filter paper. The exact process was repeated twice to ensure better activation results. Additionally, the material was placed in an oven at 353 K for 24 h to obtain reddish crystals (Fe III) [8]. Finally, they were stored in a desiccator. 

### 2.6. Characterization of the Materials

The materials were characterized by X-ray diffraction (XRD) in an X’Pert diffractometer (Philips, Amsterdam, Netherlands) with a CuKα (λ = 1.54 Å) radiation source, with a step size of 0.02° in 2θ per second in the range of 5–50° in 2θ, 45 kV, and 40 mA. Fourier transform infrared (FTIR) spectra were determined in a Magna-IR 750 device (Thermo Nicolet, Champaign, IL, USA) by the KBr pellet technique, using controlled amounts of KBr (ratio of 1 mg sample to 100 mg KBr) to address the intensity of comparative bands between the samples. Raman spectra were recorded by an inVia microscope (Renishaw, Gloucestershire, UK), using a green laser (λ = 532 nm) as the excitation line, 1% laser power, and a measurement range from 100 to 2000 cm^−1^. N_2_ adsorption-desorption isotherm at 77 K was performed in a BELSORP-Max instrument (BEL JAPAN Inc., Montgomeryville and York, PA, USA). Before analysis, samples were degassed at 403 K for 24 h in nitrogen flow to remove water and physisorbed gases. The materials’ zero charge point (ZCP) was also determined at 25 °C for 180 min in a batch system by different NaCl concentrations in 50 mL and 15 g for each MOF, adjusting the pH from 2.0 to 14.0 by addition of 0.1 mol/L HNO_3_ or NaOH. We obtained the zero charge point through the initial-final pH plots versus the initial pH [31].

### 2.7. Evaluation of the Materials

Before analyzing the samples, calibration curves were determined by a Varian 100G UV-Vis spectrometer (Agilent Technologies, Mulgrave, Australia) using the Lambert–Beer law (Equation (1)) [32]:(1)$$A_{\lambda }=\varepsilon_{\lambda }\;d\;$$
where *C* is the concentration (mg/L), *A_λ_* is the absorbance at a specific wavelength; d is the thickness of the sample contained in the cell (cm), and *ε_λ_* is the absorptivity constant ((mg/L)^−1^ cm^−1^).

We prepared mother solutions of 50 ppm of diclofenac sodium (DCF, C_14_H_10_C_l2_NNaO_2_, 98%), 15 ppm of naproxen sodium (NS, C_14_H_13_NaO_3_, 98%), and 21 ppm of ibuprofen (IB, C_13_H_18_O_2_, 98%). Then, we prepared another six solutions of different concentrations to determine the three compounds’ calibration curves.

The adsorbed quantity in the materials was calculated from the mass balance with Equation (2), where *C_0_* and *C_t_* (mg/L) are the liquid-phase concentrations of the pollutant in *t* = 0 and *t* = t_n_, respectively, and *V* (L) and *W* (g) are the volumes of the solution and the mass of the adsorbent, respectively.
(2)$$qt=\left(C0-Ct\right)\;\frac{V}{W}$$

For the adsorption tests, 3 mg of adsorbent was added to 50 mL of the initial concentration of the drug (50 ppm for DCF, 20 ppm for NS, and 15 ppm for IB) at pH values of 4.5 (DCF), 5.6 (NS), and 3.5 (IB) and a temperature of 303 K. After, aliquots were taken at certain adsorption times (10, 15, 30, 60, 120, 240, and 360 min). They were analyzed in the UV-vis spectrometer. The contact time of the material with the contaminant was 6 h. The final concentration of aliquots was determined using the maximum absorption band at a wavelength of 276 nm for DCF, 230 nm for NS, and 221 nm for IB. For obtention of the adsorption isotherms, six concentrations of each adsorbate were prepared (5, 10, 20, 30, 40, and 50 ppm for DCF; 2, 3, 6, 9, 12, and 15 ppm for NS; and 2, 4, 8, 12, 16, and 20 ppm for IB). These concentrations were selected because they are the maximum solubility of each drug, and when working with these quantities, we will obtain the maximum adsorption capacity concerning this variable.

We prepared a multicomponent aqueous mixture with 50, 15, and 20 ppm of DCF, NS, and IB, respectively, at pH 5.6. After adsorption of drugs in the adsorbents, the compounds were recovered using an aqueous solution at pH 7 at 308 K in vigorous stirring for 2 h.

The recovery of the drugs was carried out in a batch system at ambient conditions. An amount of 3 mg of the material with the drugs adsorbed was prepared in a solution of 50 mL of deionized water at 303 K and pH = 7, stirring for 30 min. Finally, the material was filtered and the concentration of the re-cured drugs in the filtering water was determined by UV-vis spectroscopy.

## 3. Results

### 3.1. Material Characterization

The magnetite nanoparticles had a crystal size of 19.5 nm, measured by the Scherrer equation. The average particle size was 90 nm, measured by scanning electron microscopy (see the magnetite’s characterization in the supporting information section).

XRD obtained the structures of pristine FeBTC and composite material (Figure 2a). An X-ray diffraction pattern characteristic of pristine FeBTC was obtained, consistent with what was reported in the literature [33]. The diffracting patterns’ noise was attributed to the weak signal from the diffraction of the peaks, caused by the fact that it is a semi-crystalline material with microcrystallinity. Those aberrations were observed in this work and various reports on this material [6]. In the diffraction pattern of the Fe_3_O_4_–FeBTC composite, peaks of pristine FeBTC were observed, with two main peaks at 2θ = 36 and 44°, which were assigned to the magnetite nanoparticles [34,35,36] according to the database (JCPDS 19-0629). Incorporating Fe_3_O_4_ nanoparticles in the FeBTC did not result in any considerable modification of the MOF’s structure.

FTIR spectra of pristine FeBTC and composite material are shown in Figure 2b. In general, the bands observed corresponding to the organic ligand (trimesic acid) of the MOF FeBTC spectrum show characteristic bands of the MOFs based on bencentricarboxylate. Several leading bands are shown in the 1508–1623 cm^−1^ region, associated with asymmetric stretching vibrations of BTC carboxylate groups. The top bands at 1384 and 1405 cm^−1^ are typical of symmetrical stretching vibrations of the same group of bands shown at around 1000 cm^−1^, representing the C–C vibrational groups, and about 760 cm^−1^, meaning the C–H bonds of the aromatic ring [37]. A top band at 938 cm^−1^ was observed, corresponding to the metal interacting with carboxylate groups [38]. Other bands observed at 430, 580, and 635 cm^−1^ [37], assigned to symmetrical and asymmetrical vibrations of the Fe–O bond of the iron oxo-cluster. The Fe (II) cations can be identified through the top band located at 430 cm^−1^ and the Fe (III) with top bands at 580 and 635 cm^−1^ [39]. These same bands were detected in the spectrum of the Fe_3_O_4_–FeBTC composite.

Additionally, a band at 1250 cm^−1^ was observed. Sudman et al. [40] showed that this band could indicate magnetite presence and refers to C–O bonding derived from a possible interaction between carboxyl groups and magnetite. The band gives further evidence of magnetite presence at around 1600 cm^−1^, which presents slight unfolding and is typical of the spectrum of magnetite [41].

Raman spectroscopy was used to determine the structure and presence of magnetite in the composite material. Figure 2c shows the Raman spectra of FeBTC and composite material. Two critical zones were observed in the Raman spectrum: the first zone between 1800 and 730 cm^−1^ is associated with vibrations of bencentricarboxylate organic ligand, and the second one at less than 600 cm^−1^ corresponds to the presence of metal coordination with carboxylate groups. More specifically, in the 1612–1003 cm^−1^ range, vibrations were observed related to the aromatic part of the MOF upon the presence of C=C. The peaks at 826 and 742 cm^−1^ are typical of C–H group stretching and bending vibrations. The peaks located at 1461 and 1546 cm^−1^ are related to symmetrical and asymmetrical carboxyl groups’ asymmetrical vibrations, respectively [42].

On the other hand, the zone associated with the metallic part has certain variations concerning the positioning of the bands related to O–M coordination; they vary between 118 and 500 cm^−1^. Finally, the bands around 180 cm^−1^ relate to the M–M interaction of the exposed metallic site or “open metal site” [43]. In the Raman spectrum of the composite material, broadband at around 670 cm^−1^ and magnetite was observed.

The textural properties of FeBTC and composite material were obtained using N_2_-physisorption at 77 K, and the surface area was acquired by the Brunauer–Emmett–Teller (BET) method. According to the IUPAC classification, the results (Figure 2d) show a type II isotherm characteristic of mesoporous materials [44]. This result indicates preferential adsorption in the monolayer up to a certain pressure—in this case, around 0.1 relative pressure (point B); above this pressure, a multilayer is formed with lower affinity. The curve above the adsorption process (in the direction of relative pressure to P_0_ = 1) is the desorption curve (if the relative pressure to P_0_ = 0), which returns by a different path due to the pores’ capillary condensation material. This phenomenon is called a hysteresis loop, and its shape can give us information about geometry and pore type. According to the IUPAC classification, these materials also show small amounts of hysteresis type H4, confirming the presence of mesoporous materials, which correspond to pores shaped like an inkwell—spherical with a narrow neck. Decreases in the BET surface area and pore volume compared to pristine FeBTC were observed for the composite material. The reduction in surface area of Fe_3_O_4_–FeBTC compared with pristine FeBTC could have been due to the obstruction of pores by the magnetite nanoparticles or the generation of surface defects from the inclusion of magnetic material; regardless, the pore size increased, and its volume decreased (see Table 1).

The pH of zero charge point (ZCP) is an essential signaler of the adsorbent’s surface charge and its preference to ionic species (Figure 3). According to the values obtained (ZCP = 3.2 and 4.4, for FeBTC and Fe_3_O_4_–FeBTC, respectively), when the pH is lower than these values, the material has positively charged sites predominantly sorbs anions, and values of greater importance because a negatively charged surface possesses them.

### 3.2. Adsorption of Pollutants on Materials 

It has been observed that the adsorption of contaminants using only magnetite [45,46] is significantly lower than that of MOFs. In the same way, several researchers who incorporated magnetite into various materials observed a considerable increase in drug removal in an aqueous environment, and consistently achieved abysmal results in adsorption in pure magnetite [47]. Therefore, the study of adsorption of pure magnetite is omitted.

#### 3.2.1. Adsorption Kinetics

Generally, the kinetic models that best describe the adsorption process in carbon materials are pseudo-first-order (Equation (3)) or pseudo-second-order (Equation (4)) equations, and sometimes the Elovich equation (Equation (5)) [48]:(3)$$qt=qe\;\left[1-e\left(-k_{1}\;t\right)\right]$$
(4)$$qt=\frac{\left(qe^{2}\;k_{2}\;t\right)}{\left(qe\;k_{2}\;t+1\right)}$$
(5)$$qt=\frac{1}{\beta }ln\left(\alpha \beta \right)+\frac{1}{\beta }ln\left(t\right)$$
where: *t* (min) is a specific period; *qt* and *qe* (mg/g) is the drug amount adsorbed at time *t*, and the equilibrium, respectively; *k*_1_ (min^−1^) and *k*_2_ (g/mg min^−1^) are the adsorption constants for the pseudo-first-order and pseudo-second-order; α (mg/g min^−1^), and *β* (g/mg) are the Elovich constants.

The kinetic adsorption model of NSAIDs in FeBTC and Fe_3_O_4_–FeBTC (Figure 4) fits a pseudo-second-order model, with a suitable correlation coefficient for pollutant adsorption (Table 2). The average time for the equilibrium adsorption was 240 min, which is similar to the reported results [11,12].

The constant rate k_2_ shows how fast the adsorption process of each drug in the materials reaches equilibrium; the higher this coefficient, the higher the adsorption speed. In general, ibuprofen is absorbed faster, and naproxen takes the longest to reach equilibrium. FeBTC reaches equilibrium slightly earlier than the composite material, which could be attributed to magnetite pores’ clogging [49,50].

#### 3.2.2. Adsorption Isotherms

The adsorption models were used to calculate the theoretical adsorption from the experimental data since the parameters obtained had so much importance in evaluating the adsorbent’s effectiveness. The most representative models that usually best describe the process of NSAID adsorption on MOFs are the Langmuir (Equation (6)), Freundlich (Equation (7)), Temkin (Equation (8)), and Dubinin–Radushkevich (Equation (9)) models: (6)$$qe=\frac{qm\;kL\;Ce}{1+kL\;Ce\;}\;$$
(7)$$qe=k_{F}\;Ce^{\left(1/n\right)}$$
(8)$$qe=A\;ln\left(Ce\;k_{T}\right)$$
(9)$$qe=qm\;exp\;\left[k_{DR}\;{\left[ln\left(1+1/Ce\right)\right]}^{2}\right]$$
where: *Ce* (mg/L) represents the equilibrium aqueous solution concentration; *qe*, and *qm* (mg/g) is the equilibrium and maximum adsorptive capacity, respectively; *kL* (L/mg), *kF*, *kT*, and *kDR* are the Langmuir, Freundlich, Temkin, and Dubinin–Radushkevich adsorption constants, respectively, which also have thermodynamic information about the affinity adsorbent-adsorbate, and n and A are dimensionless constants.

Figure 5 shows the adsorption isotherms for FeBTC and Fe_3_O_4_–FeBTC for DCF, NS, and IB, adjusted for the different adsorption models. Table 3 shows the parameters obtained from the isotherms of pollutants in FeBTC and Fe_3_O_4_–FeBTC MOFs. DCF adsorption in the materials adjusted better to the Langmuir model, indicating the presence of an energetically homogeneous surface during the adsorption process. On the other hand, there was minimal interaction between DCF molecules, suggesting that the adsorption process just forms a monolayer, which is in good agreement with previous reports [11,49,50,51,52].

The naproxen adsorption process of the MOF is well depicted by the Freundlich model using FeBTC material and the Temkin model for Fe_3_O_4_–FeBTC composite. In this case, the adsorption energy decreased while the naproxen molecule’s adsorption increased; therefore, the highest energy sites are on the material modified surface, without adsorbate–adsorbate or weak interactions [49,50].

Finally, ibuprofen adsorption in the FeBTC is better described by the Langmuir model. However, when the magnetite is incorporated, the experimental data are better described with the Freundlich model, probably due to an energetic modification after adding the magnetic material, such as a reported result [53].

The FeBTC maximum adsorption capacity for NSAID pollutants was increased by the influence of magnetite, as shown in Table 3. The improved adsorption could be attributed to magnetite modifying the material’s surface by adding superficial substitutional defects [54,55], providing energy to the electrostatic energy [10] that magnetite already possesses by itself and favors the interaction of bonds [56,57]. Although the composite material had a significantly smaller surface area, H-bonding possibly played a vital role in NSAID adsorption.

#### 3.2.3. Adsorption Thermodynamic Parameters 

Adsorption enthalpy (ΔH°), free energy (ΔG°), and entropy (ΔS°) can be calculated according to Equations (10) and (11) [53]: (10)$$ln\;\frac{qe}{Ce}=\frac{-\Delta H{{}^{\circ}}_{ads}}{R}T^{-1}+\frac{\Delta S{}^{\circ}}{R}$$
(11)ΔG°=ΔH°−TΔS°
where *R* is the constant of gases (8.314 J/mol K) and *T* is Kelvin’s temperature.

Adsorption tests were carried out at different temperatures (293, 298, 303, 213, and 318 K) at a maximum concentration of adsorbate (50 ppm for DFC, 15 ppm for NS, and 20 ppm for IB) in 50 mL solutions and with 3 mg of MOF adsorbent material. The obtained values of the thermodynamic properties for FeBTC and Fe_3_O_4_–FeBTC are shown in Table 4.

In all pollutant adsorption processes, the free energy change (ΔG°) is negative, which indicates that it is a spontaneous process. All values are negative for the enthalpy change (ΔH°), suggesting an exothermic process in most tests. Simultaneously, it is possible to detect lower energy values, as we can observe (ΔH° > −40 kJ/mol), which indicates a physisorption process. In the case of ibuprofen adsorption, higher energy values are obtained (ΔH° > −40 kJ/mol), which refers to a process of chemisorption [57]. The negative value of entropy indicates a decrease in the randomness in the solid-liquid interface during the adsorption process and a lower degree of freedom of the adsorbed species [58], represented by ΔS° values near zero, as described in the physisorption processes [59].

#### 3.2.4. Multicomponent Mixture Adsorption

Table 4 shows the adsorption capacity of the multicomponent mixture of DCF, NS, and IB by FeBTC and Fe_3_O_4_–FeBTC materials at 303 K and pH 5.6, and pollutant drug recovery capacity after desorption in the water at 303 K and pH 7.

In the multicomponent mixture adsorption study, the most adsorptive component was ibuprofen. Next came diclofenac, according to its maximum capacities in a simple system (122.9 and 357.1 mg g^−1^, respectively, for Fe_3_O_4_–FeBTC); naproxen adsorption in the simple system reached 70 mg g^−1^ for Fe_3_O_4_–FeBTC. The single MOF and the composite had a better affinity for ibuprofen since the adsorption heat values are the highest among the NSAIDs to the point of exceeding the physisorption limit, indicating the presence of a chemisorption process with ΔH° values up to 50 kJ/mol. This can be observed in the recovery of ibuprofen, which was below 10% for FeBTC. Adsorption close to 100% could not be expected for each drug rather than the simple system because there is competitive adsorption at the same type of site, similar to a reported work [11].

On the other hand, naproxen (NS) recovery was almost 87% in Fe_3_O_4_–FeBTC. DCF had the lowest adsorption energy compared to the other NSAIDs and therefore was expected to be the one with the highest amount recovered; however, we recovered only 85% for Fe_3_O_4_–FeBTC. We could explain these results by considering sites with higher interaction energy and greater affinity with adsorbate, and Table 5 shows the data. We analyzed how the process variables affect the MOF–drug adsorption process, studying the concentration of adsorbent, temperature, and pH.

#### 3.2.5. Effects of Adsorbent Mass, Temperature, and pH on the Adsorption Process

The adsorption capacity differences (mg_adsorbent_/g_adsorbate_) concerning the amount of adsorbent show that the lower the adsorbent/adsorbate ratio, the higher the adsorption capacity. The explanation for the increase in adsorption capacity is that the adsorbate increases the chance of contact with the adsorbent surface. This behavior occurred in both materials with all NSAIDs.

After studying the influence of temperature (Appendix A) on the adsorption process, a slight increase in the drugs’ adsorption capacity was seen when the temperature decreased, which shows the exothermic and spontaneous nature of the process.

The variable that most influences the adsorption process is pH (Appendix A); for high pH values, adsorption decreases significantly [11,12,49,50,51,52,53]. The materials’ isoelectric point can explain this (ZCP-3.2 and 4.4 for FeBTC, Fe_3_O_4_–FeBTC, respectively). Below its isoelectric point, a positive surface load density is guaranteed in materials in good interaction with the anionic functional groups of NSAIDs. However, the pH value is limited to the pKa value drugs’ solubility (4 and 4.2 for DCF and NS, respectively), which are very close to the materials’ isoelectric points, while the IB is soluble up to pH 3, providing a better pH adsorption margin [10].

#### 3.2.6. Characterization of Materials after Adsorption

After adsorption, the materials were characterized by XRD and Raman spectroscopy, as shown in Figure 6. The X-ray diffraction pattern of pristine FeBTC after adsorption showed a decrease in peak intensity. Additional peaks were observed associated with α-FeOOH [53], corresponding to the Joint Committee on Powder Diffraction Standards (JCPDS) data (JCPDS number 29-0713) (Figure 6a).

Raman spectra (Figure 6b) show that the MOF and composite structures remained completely stable after adding pollutants in an aqueous solution in the range of pH 3.5–5. The most common difference is shown in the two materials after adsorbing the NSAIDs; the band’s relative intensity associated with the metal cluster intensified. The bands at around 125 and 180 cm^−1^ separated and looked more pronounced. Besides, the bands’ shift observed at 1461 and 1546 cm^−1^ for both materials, was attributed to the bonds of the carboxylate group attached to the metal clusters and the open metal sites, which is evidence of an interaction of drugs at these sites. Another band observed with greater relative intensity at 1003 cm^−1^ associated with the C=C bonds of the aromatic part of the MOF confirms that the most critical interaction sites are the C=C bonds reported by various authors. Further, π interactions between the aromatic groups of the MOF and those that make up the NSAIDs, in addition to a strong interaction in the metal cluster translated as electrostatic interaction, are supported by the evidence shown by other researchers [11,12,27].

The only MOF that presented a significant change at 1250 and 1700 cm^−1^ was Fe_3_O_4_–FeBTC, which may be because the NSAIDs began to be adsorbed in another area due to the surface energy modification produced by the incorporation of magnetite.

Figure 7 compares the materials before the adsorption process and after the adsorption of the drugs (*). Here we can observe the appearance of small bands around 1280 and 2860 cm^−1^, which are specific to drugs, C–Cl vibrations in the aromatic ring, and stretching vibrations of the C–H groups, respectively [11,57]. Therefore, these give evidence of hydrogen bond formation between Fe_3_O_4_–FeBTC, the drugs, and π-π interaction, as observed by Raman spectroscopy.

On the other hand, in the FeBTC diffraction pattern, a structural disturbance was observed after the adsorption process, unlike Fe_3_O_4_–BTC, which maintained the same diffractogram. Although the FeBTC continued to preserve its structure, a typical diffraction pattern of the partial collapse was observed, which did not occur in the composite material since the magnetite provides stability for the MOF. The decrease in the characteristic peaks of magnetite nanoparticles in FeBTC could be due to re-dispersion or size reduction. The X-ray technique can no longer detect this since Raman and infrared spectroscopy confirmed the presence of the material. 

The results of the Raman spectroscopy analysis of NSAIDs in our previous work [12,53,54] helped us to build the model or interaction shown in Figure 7. 

#### 3.2.7. Reutilization and Economy 

The reuse process is crucial when looking for materials to remedy environmental problems; for this purpose, it is necessary to study and guarantee its efficiency in different adsorption cycles. These materials were easily regenerated using only water and modifying the pH at room temperature, in addition to recovering the drug from the adsorbent material. Figure 8 shows three reuse cycles for FeBTC and Fe_3_O_4_–FeBTC in the adsorption process of the three drugs. In the grayscale (for diclofenac adsorption), for both materials, almost constant adsorption is observed. In the blue scale (naproxen adsorption), we observed practically endless adsorption; in the green scale (ibuprofen adsorption), the adsorption followed after the first cycle decreased. The only drug with reduced adsorption efficiency (Figure 8) is ibuprofen because it is chemisorbed at some sites. The enthalpy of ibuprofen’s adsorption is such that it chemisorbs and deactivates some sites, leaving them unavailable for subsequent adsorption.

According to the values obtained, 1 g can treat more than 16 L of contaminated water at maximum amounts with NSAIDs. Regarding the values obtained from the adsorption kinetics: we could treat 200 L of water in one day. This material’s reusability makes it a good candidate for the sequestration of not only drugs but also heavy metals, inorganic compounds, colorants, etc.

## 4. Conclusions

We prepared a novel and stable composite material, FeBTC MOF, using a solvothermal method incorporating magnetite in situ. It was used for removing drug pollutants (naproxen sodium (NS), diclofenac sodium (DCF), and ibuprofen (IB)) in an aqueous medium. X-ray and Raman spectroscopy analysis confirmed the presence of magnetite in the composite material. A pseudo-second-order model best describes the drug’s sorption kinetic processes. Simultaneously, the thermodynamic study revealed that the three drugs’ adsorption was a feasible, spontaneous, and exothermic process. The comparison of Raman spectroscopy before and after the adsorption process showed a minor decay of the C–C, C–O, and O–Fe vibration bands, which indicates that the adsorption sites are carried in the open metal sites and by π-π stacking. The open metal sites, carboxylate functional groups, acid-base properties, and magnetic proprieties of the composite material not only make this material a good candidate for adsorption processes, but the same composite materials could also be applied in photocatalysis, organic catalysis, electrocatalysis, gas storage, supercapacitors, etc. For this reason, future research could be aimed at theoretically understanding and correlating the surface properties of this type of material to enhance its properties.

## Figures and Tables

**Figure 1 materials-14-02293-f001:**
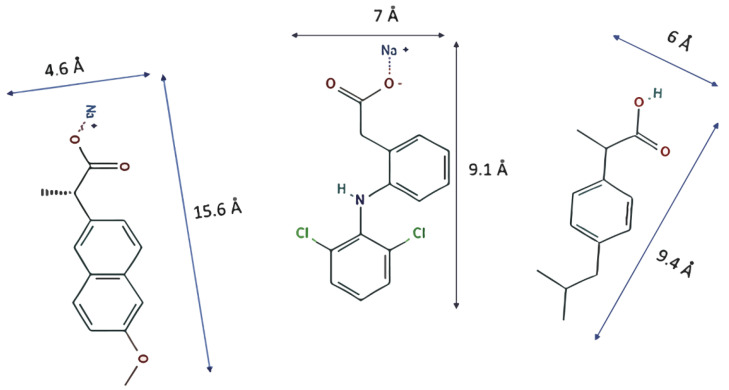
Molecules with their dimensions: naproxen sodium (**left**), diclofenac sodium (**center**), and ibuprofen (**right**).

**Figure 2 materials-14-02293-f002:**
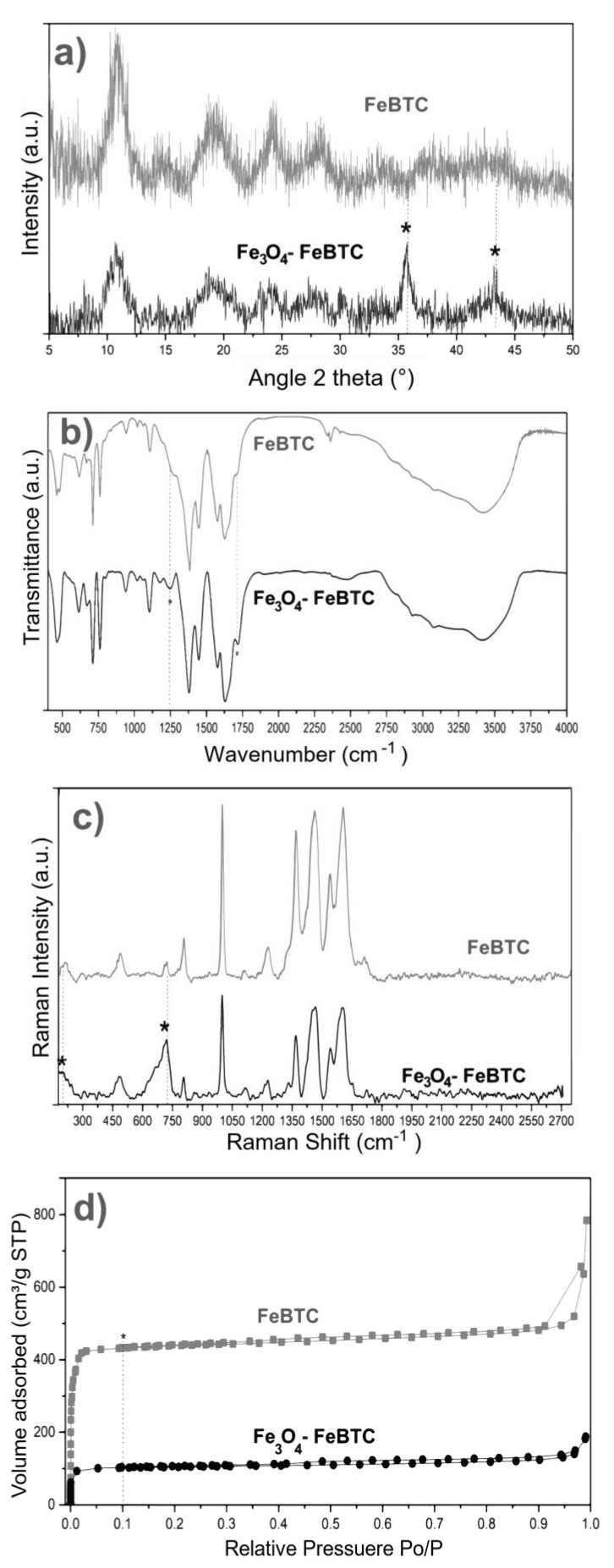
(**a**) XRD patterns, (**b**) FTIR spectra, (**c**) Raman spectra, and (**d**) N_2_ adsorption–desorption of FeBTC and Fe_3_O_4_–FeBTC at 77 K.

**Figure 3 materials-14-02293-f003:**
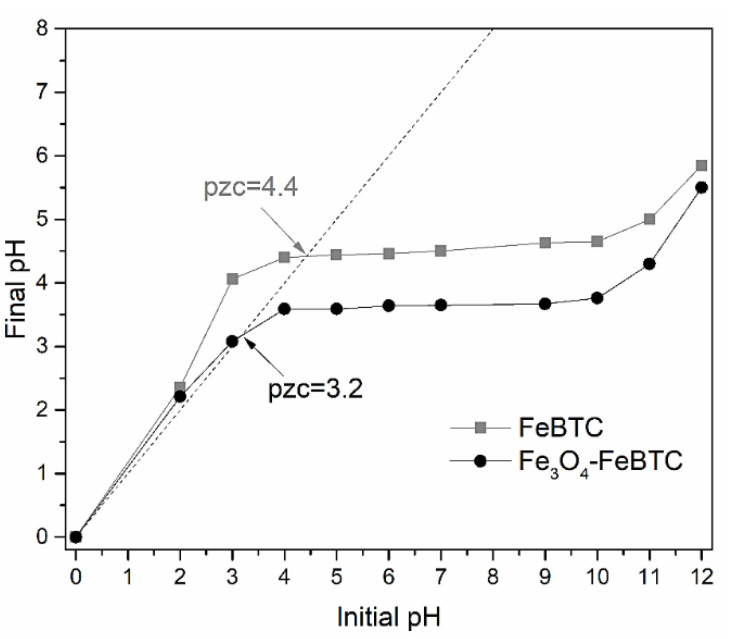
Isoelectric point determination for the FeBTC and the Fe_3_O_4_–FeBTC.

**Figure 4 materials-14-02293-f004:**
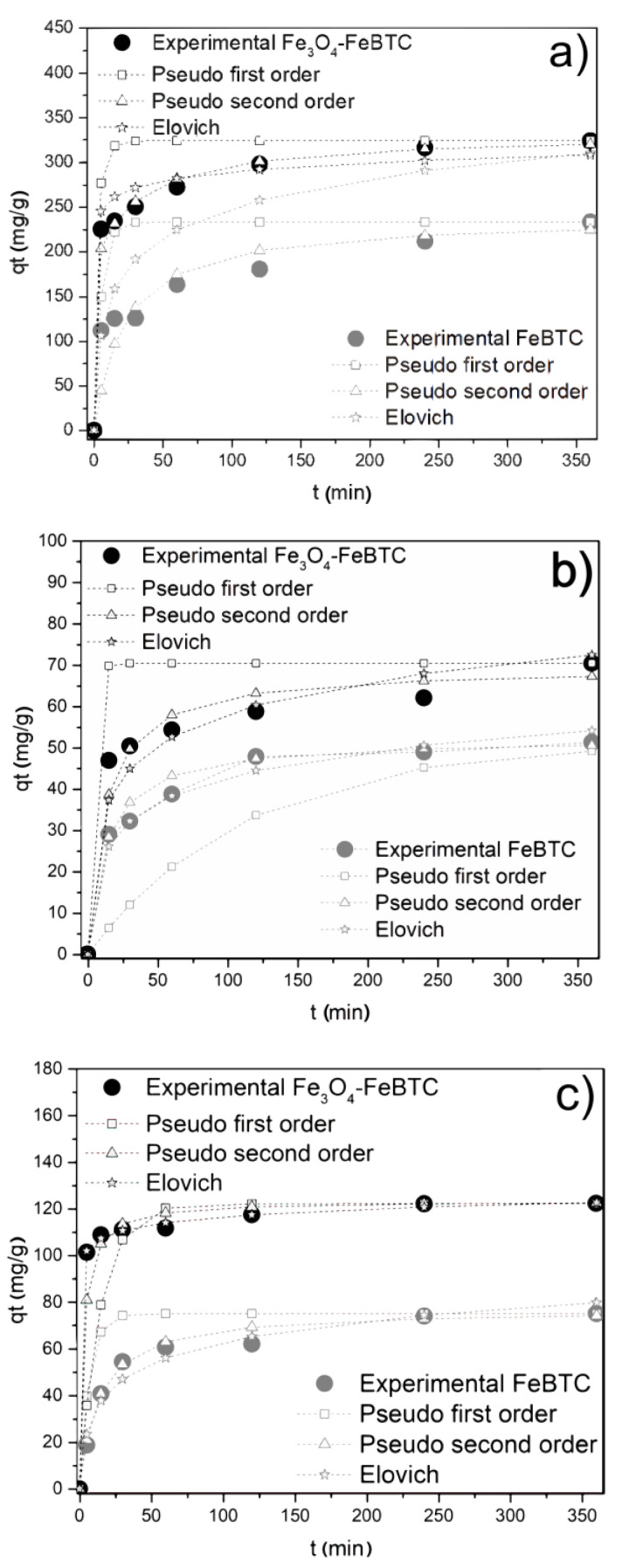
Effects of contact time on adsorption of (**a**) diclofenac sodium, (**b**) naproxen sodium, and (**c**) ibuprofen on FeBTC and Fe_3_O_4_–FeBTC by different kinetic methods. Reaction conditions: 3 mg of adsorbent; reaction temperature 303 K; pH 4.5 (DCF), 5.6 (NS), and 3.5 (IB); initial concentrations of 50 ppm (DCF), 15 ppm (NS), and 20 ppm (IB).

**Figure 5 materials-14-02293-f005:**
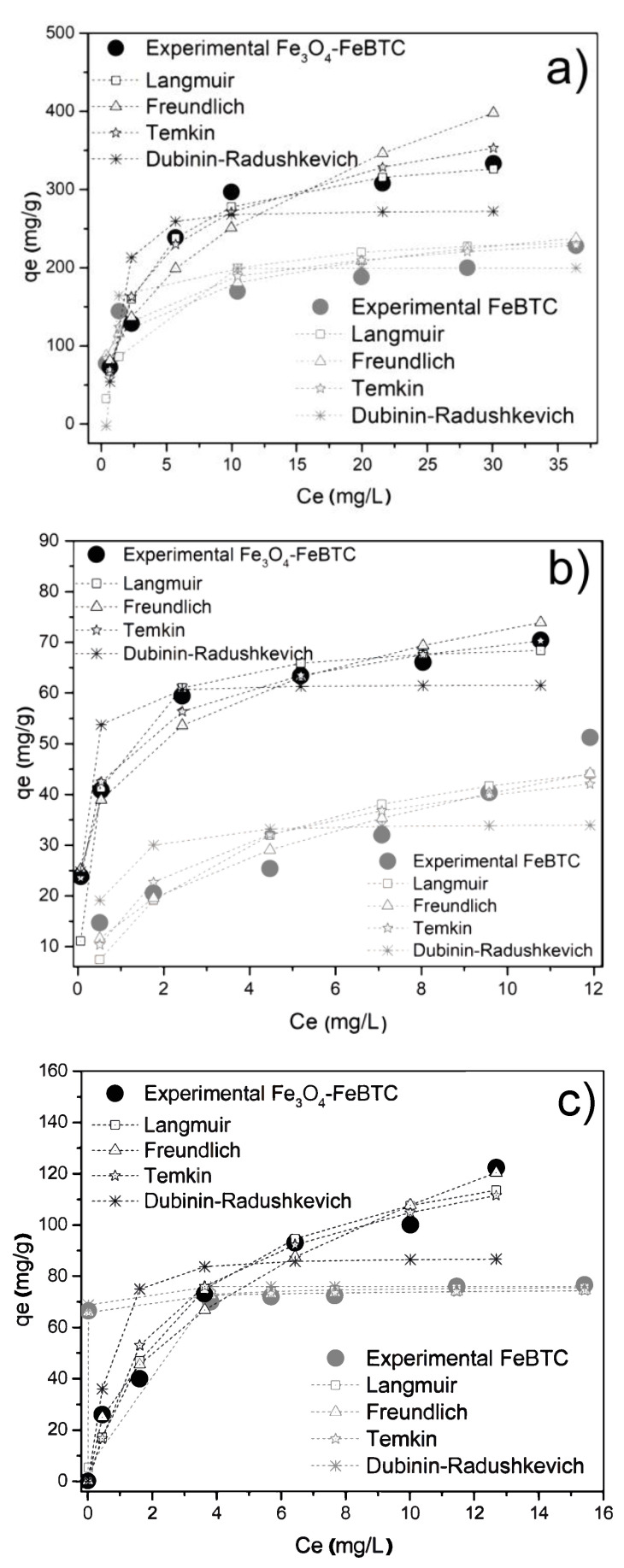
Adsorption isotherms of (**a**) diclofenac sodium, (**b**) naproxen sodium, and (**c**) ibuprofen over FeBTC and Fe_3_O_4_–FeBTC, comparing Langmuir, Freundlich, Temkin, and Dubinin–Radushkevich methods.

**Figure 6 materials-14-02293-f006:**
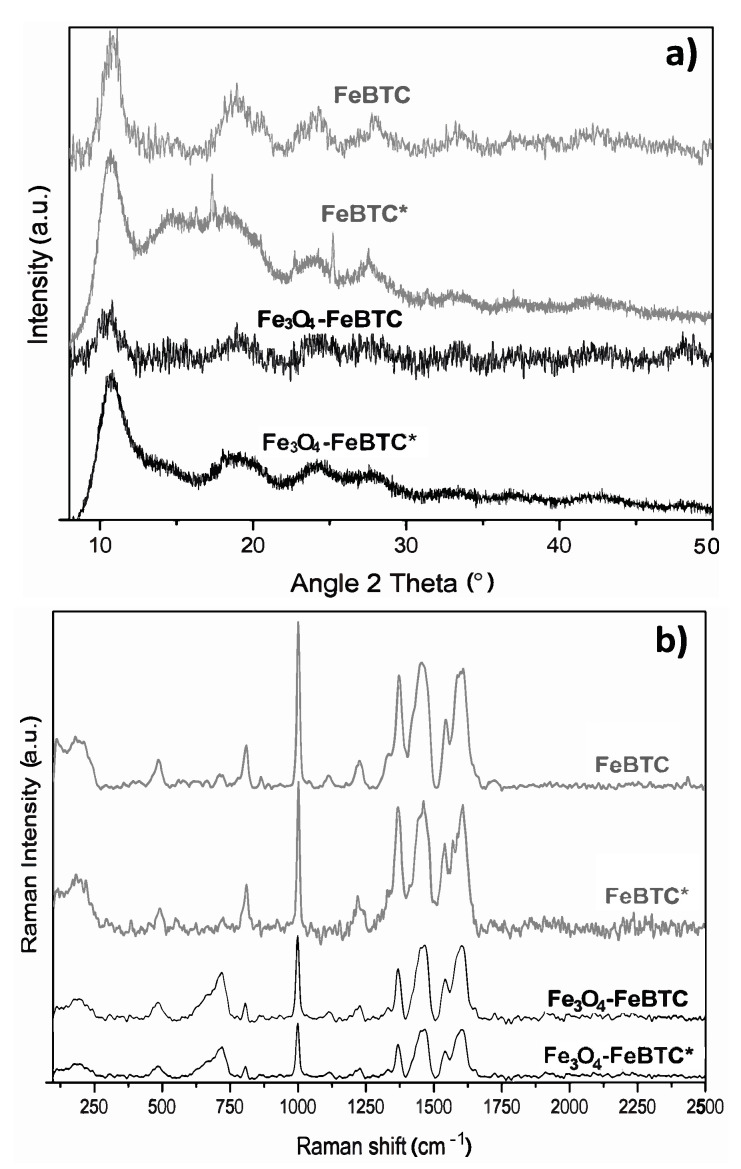
(**a**) X-ray diffractogram and (**b**) Raman spectra (**c**) of materials in their pristine state and (*) after adsorption process with pollutants.

**Figure 7 materials-14-02293-f007:**
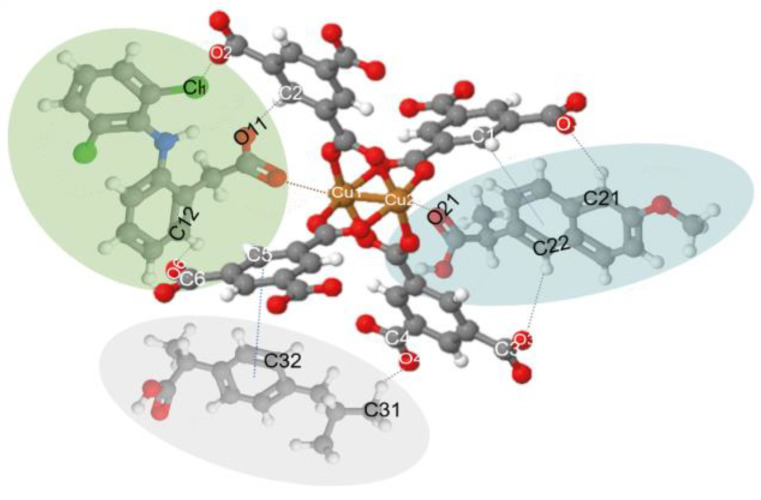
Possible interactions of drug molecules (DCF, NS, and IB) with MOF structure. (C: Gray; H: White; O: Blue; Cu: Orange; Cl: Yellow; N: Green; DCF: Green oval; NS: Blue oval; and IB: Gray oval.).

**Figure 8 materials-14-02293-f008:**
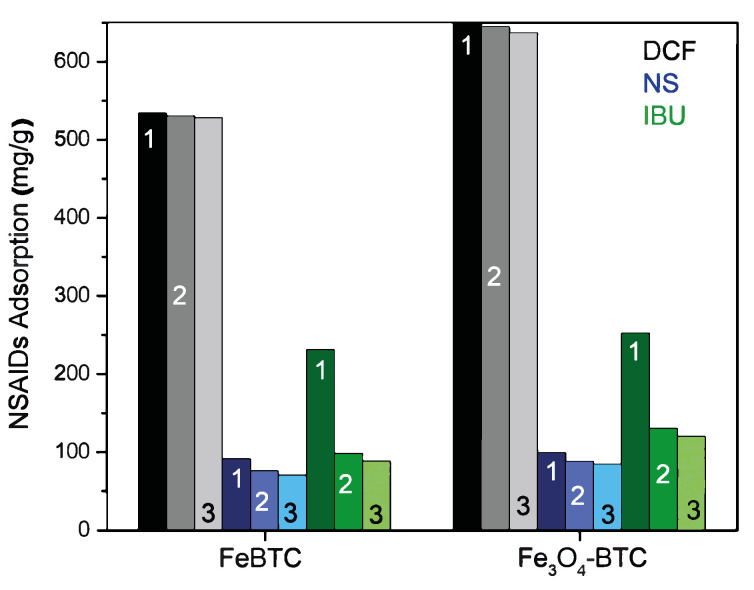
Adsorption cycles for three drugs on pristine FeBTC MOF and magnetite Fe_3_O_4_–FeBTC composite.

**Table 1 materials-14-02293-t001:** Surface area parameters of FeBTC and Fe_3_O_4_–FeBTC metal–organic frameworks (MOFs).

Material	Surface Area BET (m^2^/g)	Total Pore Volume (m^3^/g)	Pore Diameter BJH (nm)
FeBTC	815.84	0.92	3.26
Fe_3_O_4_–FeBTC	217.04	0.86	3.71

**Table 2 materials-14-02293-t002:** Kinetic parameters obtained from adsorption of contaminants in MOFs FeBTC and Fe_3_O_4_–FeBTC.

Model	Parameter	Diclofenac Sodium		Naproxen Sodium		Ibuprofen	
		FeBTC	Fe_3_O_4_–FeBTC	FeBTC	Fe_3_O_4_–FeBTC	FeBTC	Fe_3_O_4_–FeBTC
Pseudo First-Order	qt (mg/g)	139	233.1	51.2	70.4	75.1	102.6
	K_1_ (min^−1^)	0.22	0.2	0.01	0.31	0.15	0.23
	R^2^	0.994	0.981	0.920	0.962	0.950	0.920
Pseudo Second-Order	qt (mg/g)	149.3	238.1	52.4	69.4	76.9	114.9
	K_2_ (g/mg min)	9.3 × 10^−6^	1.6 × 10^−6^	8.8 × 10^−7^	3.5 × 10^−7^	2.9 × 10^−5^	2.3 × 10^−5^
	R^2^	0.995	0.991	0.997	0.990	0.990	0.940
Elovich	β (mg/g)	85.53	87.96	11.35	21.43	15.54	14.47
	α (g/mg min)	0.068	0.021	0.113	0.090	0.076	0.052
	R^2^	0.971	0.924	0.981	0.930	0.970	0.930

**Table 3 materials-14-02293-t003:** Langmuir, Freundlich, Temkin, and Dubinin–Radushkevich parameters obtained from isotherms of pollutants in FeBTC and Fe_3_O_4_–FeBTC MOFs.

Model	Parameter	Diclofenac Sodium		Naproxen Sodium		Ibuprofen	
		FeBTC	Fe_3_O_4_–FeBTC	FeBTC	Fe_3_O_4_–FeBTC	FeBTC	Fe_3_O_4_–FeBTC
Langmuir	q_m_ (mg/g)	247.5	347.1	56.8	70.9	76.3	142.9
	K_L_ (L/min)	0.40	0.35	0.29	0.25	0.54	0.30
	R^2^	0.987	0.989	0.854	0.98	0.990	0.961
Freundlich	n	4.6	2.4	4.6	2.7	5.8	2.1
	K_F_ ((g/mg) (L/mg))	108.3	96.1	44.2	17.1	70.8	36.4
	R^2^	0.904	0.911	0.956	0.970	0.750	0.979
Temkin	K_F_ (J/mol)	79.4	36.9	268.9	248.6	207.1	88.9
	A (L/mg)	34.0	3.9	16.7	5.6	21.5	4.0
	R^2^	0.910	0.903	0.804	0.990	0.730	0.941
Dubinin–Radushkevich	q_s_ (mg/g)	199.5	272.3	34.0	61.6	75.9	86.9
	K_DR_ ((mol/J)^2^)	1 × 10^−7^	3 × 10^−7^	1 × 10^−7^	2 × 10^−7^	3 × 10^−10^	1 × 10^−7^
	R^2^	0.876	0.791	0.617	0.890	0.821	0.731

**Table 4 materials-14-02293-t004:** Thermodynamic data of adsorption for drugs on FeBTC and Fe_3_O_4_–FeBTC.

Drug	Material	ΔH° (kJ/mol)	ΔS° (kJ/mol K)	ΔG° (kJ/mol)	R^2^
DCF	FeBTC	−10.26	−0.007	−6.7	0.995
	Fe_3_O_4_–FeBTC	−34.09	−0.083	−7.4	0.990
NS	FeBTC	−28.87	−0.079	−4.2	0.988
	Fe_3_O_4_–FeBTC	−45.89	−0.138	−2.8	0.991
IB	FeBTC	−75.04	−0.226	−8.2	0.996
	Fe_3_O_4_–FeBTC	−80.03	−0.240	−9.0	0.994

**Table 5 materials-14-02293-t005:** The adsorption capacity of multicomponent mixture and recovery of nonsteroidal anti-inflammatory drugs.

Material	Multicomponent Drug Adsorption (mg/g)			Multicomponent Drug Recovery (mg/g)		
	DCF	NS	IB	DCF	NS	IB
FeBTC	162.5	18.7	102.7	146.3 (90%)	16.4 (87%)	13.3 (13%)
Fe_3_O_4_–FeBTC	204.1	18.2	117.9	124.7 (61%)	15.6 (85%)	12.6 (11%)

## Data Availability

The study did not report any data.

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
