# Peer review of "Selective Adsorption of Aqueous Diclofenac Sodium, Naproxen Sodium, and Ibuprofen Using a Stable Fe3O4–FeBTC Metal–Organic Framework"

_materials, 2021, doi:10.3390/ma14092293_

Round 1
Reviewer 1 Report
The authors investigated, in this work, the adsorption of the nonsteroidal anti-inflammatory drugs onto a composite material, which is developed by incorporation of magnetite particles into iron–benzenetricarboxylate (FeBTC). The narrative is well-written especially the introduction section and the language is quite good except for some typographical mistakes at someplaces. However, the novelty of this work is extremely low. The synthesis of composite material is mimicked from the literature. In addition, I have other concerns of substantial nature
- It is unclear how the adsorption performance of the prepared composite was improved when the magnetite and FeBTC have low affinity to studied drugs.
- Determination of the particle size using SEM is not a good approach at all as one particle at a particle is measured, therefore the accuracy is believed to be very low.
- The improved efficiency of the material is not that significant especially for Naproxen Sodium
- The authors claim the selective adsorption of studied adsorbates. However, in a multicomponent system, the adsorption capacity of the material (Table 5) is even low than that of the unmodified material (FeBTC)
Minor comments:
- The point of zero charge, which is an important parameter and plays a critical role in the adsorption process, is not determined.
- The effect of temperature is discussed in section 3.2.5 but the data is not shown
- The recovery of the adsorbates is discussed in “Results and discussion” section, However, the methodology is lacking in “Materials and methods” section.
Taken into account the novelty and above comments, this article shall be rejected.
Author Response
a

Reviewer 2 Report
Title. Please replace Fe3O4 with Fe3O4.
According to the Instructions for Authors of the “Materials” journal: “Abstract: The abstract should be a total of about 200 words maximum. The abstract should be a single paragraph and should follow the style of structured abstracts, but without headings: 1) Background: Place the question addressed in a broad context and highlight the purpose of the study; 2) Methods: Describe briefly the main methods or treatments applied. Include any relevant preregistration numbers, and species and strains of any animals used. 3) Results: Summarize the article's main findings; and 4) Conclusion: Indicate the main conclusions or interpretations. The abstract should be an objective representation of the article: it must not contain results which are not presented and substantiated in the main text and should not exaggerate the main conclusions.” Please apply.
Conclusions. This section should be considerably improved/ shorten. The conclusion should summarize the manuscript's results and their importance, discuss ambiguous data and recommend further research.
I consider that the article can be accepted for publication only after a major revision.
Author Response
Thank you for your suggestions and comments, I have attached the improvements to the document.

Round 2
Reviewer 2 Report
The manuscript can be accepted for publication.
Minor comments: 1. first time when the authors use an abbreviation in the text, the authors should present both the spelled-out version and the short form (example the characterization techniques in abstract). 2. please uniformize XRD or PXRD (line 181).
Author Response
1.- Were included at the first time both the spelled-out version and the abbreviation
2.- The abbreviation XDR was standardized throughout the manuscript.
This manuscript is a resubmission of an earlier submission. The following is a list of the peer review reports and author responses from that submission.
Round 1
Reviewer 1 Report
The manuscript formatting does not follow the MDPI requirements. Example: many free spaces, figures and text outside the border, the plots are not of the same size, some figures contains a substantial amount of text which is too small to read, etc.
Introduction. The authors should improve/ reformulate the introduction section (literature review) in order to present (RECENT) the state of the art in the field of manuscript, while keeping only the relevant parts. Please use relevant current references. Figure 1 should be removed. The novelty must be specially highlighted to attract more readers. Also, please sharpen the description of the novelty factor in the "in this work" section of the introduction. not.
Results and Discussion. In general, the quality of figures should be considerable improved/ captions and labels could be confusing, author should revise them. The authors should introduce some studies related to their work and to correlate the obtained results according to previous/ similar studies. The obtained results should be detailed in figure, text or table, avoiding the overlapping of information.
Conclusions. Please reformulate this section in order to restate the major findings, tell the reader the contribution of this study to the existing literature, state future directions for research, etc.
The language of the manuscript has to be considerably improved. The current version of text contains a number of language issues.
Finally, I think that the paper is not proper for publication in the present format and must be "Reject".
Reviewer 2 Report
The manuscript "Selective adsorption of aqueous diclofenac sodium, naproxen sodium and ibuprofen using a stable Fe3O4-FeBTC metal organic framework" solves an interesting topic and has interesting findings. However, it requires a major revision before publication.
The language of the manuscript must be significantly improved so that it is easy to read. You need to correct the grammar. Please go through the entire manuscript and shorten and correct some sentences.
Abstract: Include your recommendations and future prospects. It is necessary to extend the abstract with the most significant results.
Please be sure that your manuscript thoroughly establishes how this work is fundamentally novel. Specific comparisons should be made to previously published materials that have a similar purpose. Please present a strong case for how this work is a major advance. This needs to be done in the manuscript itself, not just in the response to review comments.
Please be sure that your abstract and your Conclusions section not only summarize the key findings of your work but also explain the specific ways in which this work fundamentally advances the field relative to prior literature.
The significance of this study should be more emphasize in the introduction.
See these papers, which can help you. https://www.sciencedirect.com/science/article/abs/pii/S0045653518319301
https://www.sciencedirect.com/science/article/pii/S1878535218300510
Line 49, section wastewater: This statement is confirmed by this important document, which dealt in detail with wastewater. Therefore, it needs to be added here as a reference. https://www.sciencedirect.com/science/article/abs/pii/S0304389420316149
Line 77: Improve of quality figure 1.
Line 101: Check the correct use of SI units throughout the manuscript.
Line 124: You indicate that the material has been cleaned. In the materials section, you stated that all chemicals are of high purity. I wonder why there was further purification after the synthesis?
Line 132: Specify into manuscript step of shifting angles per unit time.
Line 134: ,,pellet method,, change to ,,KBr pellet technique,,
Line 135: What is the controlled amount of KBr? You must enter the exact ratio in the manuscript. Usually a ratio of 1 mg sample to 200 mg KBr is used, i.e. 1: 200. However, it depends on the specificity of the material.
Line 144: Improve the notation of equations with explanations throughout the manuscript.
Line 192: Figure 2a: What do you attribute the noise in the diffractograms to?
Line 199:,,760 cm-1 to 199 the C-H bonds of the aromatic ring,, this is already confirmed by this important paper and must therefore be added to this place. https://www.sciencedirect.com/science/article/abs/pii/S0169131719301413
Line 226-238: Better describe the hysteresis phenomenon as well as the direction of adsorption and desorption.
Line 240: Explain why the BET surface is many times larger for FeBTC 4 than for Fe3O4-FeBTC.
Line 262: Resize figure 3 to make it easy to read.
Line 266. The quality of Table 2 needs to be significantly improved.
Line 302: Enlarge image size 4 so that it is clearly visible.
Line 312: The presentation of Table 3 needs to be improved.
Line 324: I am interested in measurement deviations. What were the deviations? How many times the measurement was performed for each parameter.
Line 374: List options at higher pH.
Line 416: For better transparency, it would be best to add abbreviations of individual atoms to Figure 6.
Line 430: The quality and interpretation of Figure 7 should be improved.
Line 431: ,,Conclusion,, change to ,,Conclusions,, Extend the conclusion more with all your most important findings. Indicate the possible risks of such research. Add your recommendations for future research.
Make sure the references are added correctly according to the journal's instructions.
Line 469: Figure S1: I really like this interpretation and idea. If it were possible this image would be useful as a graphic abstract. however, it would be necessary to illustrate how drug adsorption will affect the smile of fish.
Reviewer 3 Report
In this study a new combination of iron containing benzenetricarboxylate (BTC) MOF has been used to explore removal of frequently used drug- organic contaminants from waste water. The study is in line of the intensive development of materials for pollution treatment and presents a possibly interesting step forward in exploring of potentially useful materials. Therefore, as far as no serious criticism is done by fellow reviewers, I can recommend it for publication, after correcting for some minor issues.
Minor corrections
Check English throughout the text.
line 21 – confusing
line 106 – not clear
line 237 Table with capital T
line 272-273 This sentence is misleading, as far as these isotherms are not characteristic for carbon base materials. So the sentence can be deleted, together with the cited reference. Or it can be expanded, mentioning all other types of mesoporous adsorbents suitable for organic pollutant or metal ion removal.
line 275-278 These equations were probably not used in this study, since authors report the isotherms and the fitted models in their original, non-linearized form (Figure 4) . Hence, the original equations can be shown instead.
line 367 -”antagonistic” is a not appropriate term for these observations, since the two variables do not belong to the same class.
line 371-372 sentence is not very clear.
line 383-384 Authors observe narrow peaks on the materials after adsorption (Figure 5) and hypothesize them as iron oxide species. Such assumption is not elaborated and not proven here, therefore it can be misleading, and is better to be avoided. Or, authors can try to match the new lines with one of the iron oxide phases.
Further, it is not stated, which materials are there: after processing with which adsorbate.
lines 453-455 “The surface heterogeneity for most drugs is observed by the Freundlich adsorption model, indicating adsorption in the monolayer, energetically heterogeneous adsorption surface, and a little interaction between adsorbate molecules.”
Authors apparently miss the point: The Freundlich model is largely empirical, and an experimental Freundlich behavior does not prove the monolayer type adsorption, even though such site binding energies can be proposed which lead to Freundlich behavior.
Monolayer can be concluded only from a Langmuir type experimental isotherm.
In addition, this text is more or less the repetition of the preceding paragraph.
Table 1 translate to English
Table 3 Consider reformatting the second line of the table.
line 434 give English names
line 547-548, avoid using personal form, or make it clear, who is the person.
Modify the references according to the journal style
Round 2
Reviewer 1 Report
I consider that the article can be accepted for publication only after a major revision considering that the authors did not responded to my (all) previous recommendations.
- The manuscript formatting does not follow the MDPI requirements (pdf format). Example: free spaces, figures and text outside the border, the plots are not of the same size, some figures contain a substantial amount of text which is too small to read, etc.
- Results and Discussion. The authors should introduce some studies related to their work and to correlate the obtained results according to previous/ similar studies.
I consider that the manuscript requires a carefully, major revision before resubmitting. If the manuscript will not be considerable improved, I will not recommend its publication.
Author Response
we corrected the design and presentation of the research to adapt it to the MPDI manuscript format
We introduce some studies related to their work and to correlate the obtained results according to similar studies.
Reviewer 2 Report
Dear authors.
Your manuscript has been significantly improved.
I noticed only one typo in line 601 figure S1. ,,H-bon interaction,, you probably meant ,,H-bond interaction,, if so, correct it according to my suggestion.
Furthermore, this manuscript can be accepted as a publication for the Materials journal.
Well done! I wish you all the best in further research.
Author Response
the word in figure S1 was corrected.
we also worked on the design and presentation of the research